# Adverse Childhood Experiences Predict Common Neurodevelopmental and Behavioral Health Conditions among U.S. Children

**DOI:** 10.3390/children8090761

**Published:** 2021-08-31

**Authors:** Kasra Zarei, Guifeng Xu, Bridget Zimmerman, Michele Giannotti, Lane Strathearn

**Affiliations:** 1Center for Disabilities and Development, University of Iowa Stead Family Children’s Hospital, Iowa City, IA 52242, USA; kasra-zarei@uiowa.edu; 2Department of Epidemiology, University of Iowa College of Public Health, Iowa City, IA 52242, USA; xguifeng365@163.com; 3Department of Biostatistics, University of Iowa College of Public Health, Iowa City, IA 52242, USA; bridget-zimmerman@uiowa.edu; 4Department of Psychology and Cognitive Science, University of Trento, Via Calepina, 14, 38122 Trento, Italy; michele.giannotti@unitn.it; 5Stead Family Department of Pediatrics, University of Iowa Carver College of Medicine, Iowa City, IA 52242, USA; 6Hawkeye Intellectual and Developmental Disabilities Research Center (Hawk-IDDRC), University of Iowa, Iowa City, IA 52242, USA

**Keywords:** childhood adversity, trauma, adverse childhood experiences, neurodevelopment, behavioral health

## Abstract

Objective: Adverse childhood experiences (ACEs) can have a significant but variable effect on childhood neurodevelopment. The purpose of this study was to quantify and compare the associations between “household challenge” ACEs and common childhood neurodevelopmental and behavioral health conditions, using nationally representative U.S. data. Method: This study used data from the 2016–2019 National Survey of Children’s Health, a nationwide, population-based, cross-sectional survey. Seven household challenge ACEs (not including child maltreatment) were reported by parents/guardians: parental death, incarceration, divorce/separation, family violence, mental illness, substance abuse, and poverty. Logistic regression with sample weights was used to estimate the odds ratio (OR) for 15 parent-reported neurodevelopmental and behavioral health conditions, by the number of reported ACEs. A dose-response relationship was examined by applying tests of orthogonal polynomial contrasts to fitted logistic regression models. Results: Down syndrome, Tourette syndrome and cerebral palsy were not associated with household challenge ACEs, whereas behavior/conduct problems, depression, and substance abuse were strongly associated, with adjusted ORs ranging from 6.36 (95% confidence interval (CI) 5.53, 7.32) to 9.19 (95% CI 7.79, 10.84). Other neurodevelopmental conditions not traditionally associated with childhood adversity showed moderate yet robust associations with ACEs, including autism (adjusted OR 2.15, 95% CI 1.64, 2.81), learning disability (adjusted OR 3.26, 95% CI 2.80, 3.80), and attention deficit hyperactivity disorder (adjusted OR 3.95, 95% CI 3.44, 4.53). The ORs increased with the number of ACEs, showing significant positive linear trends. Conclusion: We found significant dose-dependent or cumulative associations between ACEs and multiple neurodevelopmental and behavioral conditions.

## 1. Introduction

Adverse childhood experiences (ACEs) are defined as traumatic experiences in a person’s life occurring before the age of 18 years, which are recalled by the person in adulthood. Previous studies have categorized these self-reported adverse experiences in terms of exposure to (1) childhood abuse, (2) neglect, and (3) “household challenges” or dysfunction, broadly defined by a range of chronic stressors including parental absence, parental incarceration, family mental illness, domestic violence, and societal factors such as poverty, community violence, or homelessness. A summative ACE score has been used to measure the cumulative exposure of varying adverse childhood conditions [1].

Numerous studies have shown that ACEs are associated with long-term physical health problems, including obesity, high blood pressure [2], high total cholesterol [2], cardiovascular disease [1,2,3], cancer, and higher overall mortality rates [4,5]. Furthermore, ACE exposure has been associated with a number of neurodevelopmental and behavioral health conditions, including learning disability [6], anxiety [7,8,9,10], substance use disorders [1,8,9,10,11,12,13,14,15], depression [2,8,9,10,16], post-traumatic stress disorder (PTSD) [17,18], high-risk sexual behavior [1,19], and suicide attempts [5,20,21]. This is consistent with the idea that “toxic stress” or conditions of deprivation may lead to altered activation of the glucocorticoid stress response system, cortisol production, and disruption of neuroendocrine and immune systems, which remodel neurological pathways in brain regions such as the hippocampus, amygdala, and prefrontal cortex [22].

A recent 20-year prospective study demonstrated that child abuse and neglect is associated with a range of adverse cognitive, psychological and health outcomes, including learning and attention problems, anxiety, depression, behavior problems and substance abuse, which extends into adolescence and young adulthood [23]. The current study examines associations between seven parent-reported ACEs not specifically related to child maltreatment—referred to as household challenge ACEs—and 15 common neurodevelopmental and behavioral health conditions in children, using the most up-to-date, nationally representative data available through the National Survey of Children’s Health (NSCH). Previously published studies have reported on the associations between household challenge ACEs and many of the conditions surveyed in the NSCH, including autism spectrum disorder (ASD) [24], developmental delay [25], intellectual disability (ID) [26], headaches [27,28], learning problems [6], attention deficit hyperactivity disorder (ADHD) [29,30,31], anxiety [7,8,9,10], conduct disorder [32], depression [2,8,9,10,16], and substance abuse [8,9,10,11,12,13,14,15]. Our specific goal was to compare the strength of these associations using the most up-to-date NSCH dataset, and examine dose-response relationships, to ascertain which conditions are most strongly associated with ACEs.

## 2. Materials and Methods

### 2.1. Study Population

The National Survey of Children’s Health (NSCH) is a leading national survey on children’s health in the United States, sponsored by the Maternal and Child Health Bureau. The NSCH provides rich data on multiple, intersecting aspects of children’s lives, including physical and mental health and the child’s family and social environment. The survey investigators over-sampled children 0–5 years old and children with special health care needs (such as learning, speech, and intellectual disabilities), giving a higher probability of selection compared to other children in households with 2 or more children. The National Center for Health Statistics Research Ethics Review Board approved all data collection procedures for the survey. In the present study, we analyzed data from the most recent 2016, 2017, 2018, and 2019 NSCH [33], consisting of 131,774 questionnaires.

### 2.2. Data Collection

The respondents to the NSCH survey are parents, guardians or an adult in the household who is most familiar with the child’s health and healthcare, from civilian, non-institutional households across the U.S. Households were invited to complete a short online screening questionnaire. A mailed paper-and-pencil screening questionnaire was provided if the household did not respond to the first two web-based survey invitations. The screening questionnaire asked participants to identify all children aged 0–17 years living in the household. If a child lived in the household, the participants were directed to a more detailed, age-specific topical questionnaire. Only one child per household was randomly selected to be the subject of the detailed topical questionnaire.

As part of the survey, respondents were asked questions about the child’s past and current health conditions and diagnoses, including Down syndrome, Tourette syndrome, epilepsy, speech or other language disorder, cerebral palsy, ASD, ID, severe or frequent headaches including migraine, developmental delay, learning disability, ADHD, anxiety problems, behavioral or conduct problems, depression, and substance use disorder. In this study, cases were defined as respondents who had reported ever receiving a diagnosis of the health condition for their child, rather than a current diagnosis only.

The primary independent variables were the type of ACE exposure and ACE score. ACE questions relating to child abuse and neglect were not included in the NSCH and were therefore not part of this analysis. ACE questions relating to household challenge (also referred to as “household dysfunction”) [1] included whether the child had ever had an incarcerated parent or guardian; witnessed parents or other adults physically abusing one another in the home (“slap, hit, kick, punch one another”); or lived with anyone who was mentally ill, suicidal, severely depressed, or had a drug/alcohol problem. The remaining questions were based on a review of childhood stressors by a multidisciplinary Technical Expert Panel or were adapted from the Behavioral Risk Factor Surveillance System ACE module [34]. Specifically, the survey asked about whether the child had ever experienced parental/guardian divorce/separation or death. Finally, respondents were asked how often, since the child’s birth, it has been difficult to cover basic needs such as food or housing on the family’s income (“never”, “rarely”, “somewhat often” or “very often”, with the latter two constituting a positive ACE score) (Table 1). The survey questions relating to household challenge ACEs and caregiver-reported diagnoses are included in Appendix A.

Information on the child’s age, sex, race/ethnicity, and family income was collected through a standardized questionnaire. Family income levels were classified based on the ratio of family income to poverty guidelines specific to the survey year as <1.0, 1.0–1.9, 2.0–3.9, and ≥4.0. Parent education was classified as “Less than high school”, “High school”, “Some college or associate degree,” and “College degree or higher,” with the latter two groups combined into the “Some college or higher” classification. Household smoking, low birth weight (<2500 g), and preterm birth were categorized as “Yes” or “No”.

### 2.3. Statistical Analyses

Since the data for this study were collected using a complex survey design, the survey design parameters were accounted for in the statistical analyses. SAS Survey procedures were used that incorporated survey weights (which account for oversampling), strata, and primary sampling units of the survey design in all the statistical analyses.

Chi-square tests were used to compare categorical baseline characteristics with household challenge ACEs, defined as no ACEs, one ACE (ACE = 1), two ACEs (ACE = 2), and 3 or more ACEs (ACE ≥ 3) (Table 2). Multiple logistic regression was also performed, in which the OR and 95% CI was calculated for each of the 15 neurodevelopmental and behavioral health conditions, according to each level of ACE exposure relative to no exposure (Table 3, Figure 1A). ACE counts of 3 or more were collapsed into one category, since there were only a relatively small number of respondents with 4, 5, 6, or 7 reported ACEs. While a count cut-off of 4+ ACEs is typically used in the literature, a lower cut-off was used in this analysis since several traditionally defined ACEs (e.g., child abuse and neglect) were not available to include in this composite ACE score. The analysis consisted of an unadjusted model, and a model adjusting for age, sex, race/ethnicity, family highest education level, family income level compared to poverty level, household smoking, preterm birth, and low birth weight. Preterm birth and low birth weight were included because of the increased risk of developmental delay, learning disability, ADHD and other health conditions surveyed in the NSCH [35], while also being associated with household challenge ACEs [36].

After excluding conditions with no statistical association with any ACE score greater than zero (Down syndrome, Tourette syndrome and cerebral palsy), the odds ratios were rank ordered, based on estimates from the logistic regression analysis of conditions according to the presence/absence of any ACEs (Yes/No) (Figure 1A,B). The neurodevelopmental and behavior health conditions were then clustered into groups using k-means clustering (Figure 1C). In addition, the dose-response of the log-odds of the health condition with ACE score was examined by applying tests of orthogonal polynomial (cubic, quadratic, and linear) contrast to the fitted logistic regression model using the SAS statement ‘contrast.’ For fitting the model, ACE score was not used as a continuous variable but as a categorical ordinal variable with 4 levels (0, 1, 2, ≥3), and 3 degrees of freedom. For assessing trends, we did not fit 3 separate models, but simply partitioned the 3 degrees of freedom into 3 orthogonal contrasts to test for linear, quadratic, and cubic trends. Instead of assuming equally spaced intervals, since the last category was combined, the average score of 3.8, was used for the ≥3 category. The orthogonal polynomial coefficients for ACE levels 0, 1, 2, 3.8, which were used to test for linear, quadratic and cubic trends, are included in Appendix A.

All data analyses were conducted using SAS survey procedures, SURVEYFREQ and SURVEYLOGISTIC (SAS 9.4, SAS/STAT 14.3, SAS Institute Inc., Cary, NC, USA). *p* < 0.05 was considered statistically significant. Bonferroni correction (*p* < 0.001) accounted for 3 ACE levels and 15 conditions, or 45 possible tests.

## 3. Results

Among the 131,774 children included in the 2016–2019 NSCH, 80,399 children (61.0%) did not experience any household challenge ACEs, 27,603 children (20.9%) experienced a single household challenge ACE, 10,961 children (8.3%) experienced two household challenge ACEs, 10,938 children (8.3%) experienced three or more household challenge ACEs, while 1873 children (1.42%) had missing data (Table 2). Demographic characteristics differed among the children in the ACE groups, with those having a greater number of ACE exposures being older, having a lower level of parental education, lower family income to poverty ratio, and higher rate of household smoking (Table 2).

The logistic regression analyses, with ACEs (categorized as 1, 2 or ≥3) as independent variables, showed no significant association with Down syndrome, Tourette syndrome or cerebral palsy at any ACE level after adjusting for covariates (Table 3). For all ACE levels, after adjusting for potential confounding, there was a significant increase in odds of each of the following neurodevelopmental and behavioral health conditions: epilepsy, speech disorder, ASD, ID, headaches/migraine, developmental delay, learning disability, ADHD, anxiety problems, behavior/conduct problems, depression, and substance use disorder (Table 3). Adjusting the significance level to account for multiplicity (*p* < 0.001), all these listed conditions showed a significantly increased odds ratio for ACE ≥ 3.

Since Down syndrome, Tourette syndrome and cerebral palsy showed no significant associations with ACE scores, they were not included in the cluster analysis. Using a logistic regression based on whether any ACEs were present, the rank-order of the ORs for the remaining 12 neurodevelopmental and behavioral health conditions was calculated, as presented in Figure 1B,C. Epilepsy, speech disorder and ASD belonged to the cluster of conditions with moderate OR estimates (Figure 1(C2)). ID, severe or frequent headaches/migraines, developmental delay, learning disability, ADHD, and anxiety problems belonged to the cluster of conditions with moderately high OR estimates (Figure 1(C3)). Behavior/conduct problems, depression, and substance use disorder belonged to the cluster of conditions with the highest OR estimates (Figure 1(C4)).

A significant and positive cubic, quadratic or linear trend was found for all conditions except Down syndrome, Tourette syndrome, cerebral palsy and substance use disorder, after Bonferroni correction (*p* < 0.001) (Appendix A), and demonstrated that the odds of reporting each condition increased with the number of ACEs experienced.

## 4. Discussion

In a nationally representative sample of U.S. children with the most up-to-date data, we found a significant and positive association between modified ACE scores specific to household challenges experienced in childhood and 12 out of 15 neurodevelopmental and behavioral health conditions. There was significant variation in the magnitude of associations between ACEs and the different conditions examined. Conditions with known genetic and perinatal etiologies were not associated with ACEs whereas conditions with well-known associations with trauma and chronic stress were associated with ACEs in varying degrees. Down syndrome, Tourette syndrome, and cerebral palsy were not found to be associated with ACEs, which is consistent with our understanding of the likely etiologies of these conditions. However, most strikingly, several neurodevelopmental conditions not traditionally associated with childhood adversity were shown to have a moderate yet robust association with ACEs, including autism, epilepsy and speech disorder. Even stronger associations were seen for intellectual disability, headaches/migraine, developmental delay, learning disability, ADHD, and anxiety problems. The current study confirms other published associations between ACEs and neurodevelopmental and behavioral conditions, as well as the idea that cumulative childhood exposure to adversity may increase risk.

In addition, the current study quantifies the dose-response nature of household challenge ACEs on the odds of developing 12 prevalent health conditions among children and adolescents, and clusters them using an unsupervised approach into one of three categories: moderate, moderately high, and high risk. While other studies have shown how individual developmental and behavioral conditions are associated with ACEs, none have explored how risks associated with increasing number of ACEs vary across different health conditions, while comparing the magnitude of the associations. These results suggest a cumulative effect of household challenge ACEs on the risk of many neurodevelopmental and behavioral health conditions. Furthermore, the dose-response effects quantified in this analysis highlight the importance of examining cumulative ACE scores in the assessment of neurodevelopmental and behavioral health conditions.

Various mechanisms have been proposed to help explain this variable association between ACE exposures and neurodevelopmental and behavioral health conditions. These range from known genetic etiologies, gene variants that interact with environmental stressors to produce disease, and experiential factors driving changes in neural circuitry, possibly via epigenetic mechanisms. Neurodevelopmental and behavioral health conditions may lie along a spectrum where at one end, intrinsic factors (genetic or perinatal) predominate, as in the case of Down syndrome and cerebral palsy, and at the other end of the spectrum, extrinsic factors predominate, as in the cases of conduct and behavior problems, depression and substance use disorder (see proposed model in Figure 2). For these conditions, having three or more ACEs was associated with a 6- to 8-fold increase in odds, in adjusted models. It has long been established that these conditions are substantially influenced by environmental factors, with childhood adversity playing an important role in their etiologies [12,13,14,15,16]. On the other hand, conditions like Down syndrome, Tourette syndrome, and cerebral palsy showed no significant association with the number of household challenge ACEs, consistent with the evidence that genetic or peri-natal factors may play a more important role in their etiologies.

The category of health conditions with the moderate to moderately high strength of association reflects the larger number of conditions that may have both intrinsic and extrinsic influences. Similar associations have been reported previously for developmental delay [25], ID [26], learning problems [6], severe or frequent headaches (including migraines) [27,28], ADHD [29,30,31], and anxiety [7,8,9,10]. However, the magnitude of the cumulative effect of ACEs on risk, as well as the objective positioning of these conditions along an etiological spectrum, using the most up-to-date dataset, has not previously been reported.

It is important to note that specific ACEs (such as the parental depression vs. exposure to family violence) may have unique and differing contributions to the measured childhood outcomes. For example, while total ACE scores were associated with both internalizing and externalizing behaviors (e.g., childhood depression and behavior/conduct problems, respectively), different types of ACEs may differentially contribute to these unique outcomes. Parental depression may predict internalizing behavior problems in children, while harsh parenting and physical punishment predict externalizing behavior problems [37].

The underlying mechanisms for the observed associations between household challenge ACEs and neurodevelopmental and behavioral health conditions remains to be elucidated and ultimately cannot be further explored within this study, given limitations of understanding the temporal relationship between the study variables. One previously proposed mechanism is that chronic stress resulting from household challenge ACEs may lead to changes in brain structure and functioning [22,38]. For example, chronic stress may lead to stress sensitization, involving dysregulation of the hypothalamic–pituitary–adrenal (HPA) axis, which subsequently affects cortisol reactivity and children’s executive functioning and attention in ADHD [39]. Previous research has also explored HPA axis dysregulation and disruptions in cortisol secretion in ASD [40]. Chronic stress may attenuate serotonin (5-HT) neurotransmission and sensitivity, possibly contributing to depression and other mood disorders [41], and is associated with reduced hippocampal volume in patients with depression and PTSD [42]. As for neurodevelopmental conditions including ID, developmental delay, speech disorder, and learning disabilities, adverse childhood experiences and reduced sensory stimulation may negatively affect neural circuits and the developing brain’s architecture especially during its most plastic period in life, which can lead to cognitive impairment [43,44]. As mentioned, abuse and neglect were not included as ACEs in this study due to a lack of available data, but they are strongly associated with household dysfunction and have a negative effect on childhood development [23,45,46]. ACEs related to household dysfunction can co-occur at high frequency with maltreatment-related ACEs, and it is likely that some of the associations reported in this article—particularly the cognitive and psychological health outcomes—as well as the strength of the associations, are driven by maltreatment-related ACEs.

Another proposed explanation for these associations is that household challenge ACEs are not only stressful and detrimental for the development of children, but may adversely impact parental functioning. Previous studies have shown that challenging household conditions are associated with a higher risk for maternal depression and other mental health problems in caregivers. In this scenario, parenting sensitivity and responsiveness may be affected, leading to impaired parent-child dyadic communication, child neglect and insecure attachment styles in children [47]. A systematic review of 29 studies showed an association between insecure attachment and ADHD [48], and maternal responsiveness is a predictor of ADHD symptoms across preschool ages [49]. Maternal care is associated with increased language abilities, school readiness, cooperation, compliance, and lower rates of behavior problems [50,51], as well as fostering positive outcomes in children with ASD and developmental delay [50]. Thus, parental sensitivity and responsiveness could be an important modifiable mediator between household challenges and disorder risk.

A major strength of this study is the use of nationally representative population-based data with a large sample size, a relatively high response rate, and generalizability. Some limitations of this study merit further consideration. Firstly, cumulative risk models such as those utilized in ACE studies assume that any given exposure is equally weighted, regardless of severity, frequency or chronicity. This is not an accurate assumption and is a major limitation of the study (such as in the case of a parent with mental illness, which could range from mild post-partum depression to untreated schizophrenia). Although the cumulative risk model used in this analysis is a crude measure of early adversity compared to a child’s actual measured experience of adversity, there appeared to be a cumulative effect across different conditions within three objectively defined clusters. This is in contrast to the lack of association between ACEs and conditions such as Down Syndrome, which has a known genetic etiology but levels of disability comparable to many of the other conditions. Secondly, there is no information about when the ACEs occurred in the child’s life and when the reported diagnoses were made, which limits any temporal or causal inferences. Thirdly, all diagnoses and related information are self-reported by the parent, rather than confirmed from validated clinical records. This lack of clarity creates the possibility of misclassification of clinical outcomes, which could lead to an overestimation or underestimation of the true measure of association. Although the results were adjusted for potential confounding variables such as race and socioeconomic status, these factors may also introduce unmeasured bias into the reported differential diagnoses. Lastly, the NSCH did not provide information concerning past family history of the conditions in the survey. It is possible that a family history of neurodevelopmental and behavioral conditions may at least partially explain both the household challenges (due to increased parental stress to meet health-related needs, and resource limitations) and the child’s increased risk of these conditions. Nevertheless, the observed dose-response effects suggest otherwise.

Despite the methodological limitations, the current study found a significant and positive association between adverse childhood experiences and multiple neurodevelopmental and behavioral health conditions, even after adjusting for covariates. We have uniquely provided quantitative, ranked estimates of the specific dose-response relationships that exist between household challenge ACEs and the conditions studied. This research highlights the significant influence of household dysfunction on disease risk and affirms the need for broader-based interventions. Healthcare practitioners including physicians, psychologists, and social workers can use this data to inform their practice, particularly using individual ACE screening assessments for patients who present for the evaluation of common neurodevelopmental and behavioral health conditions analyzed in this study. The results of this study support the use of evidence-based screening and other trauma-informed practices. Furthermore, this research can be used to guide trauma-informed public health prevention efforts in the community. Although the exact mechanisms that lead to the development of these conditions remain to be understood, several converging mechanisms may play an important role in the health conditions examined. With additional longitudinal research, the mechanisms underlying these associations may be more clearly elucidated.

## Figures and Tables

**Figure 1 children-08-00761-f001:**
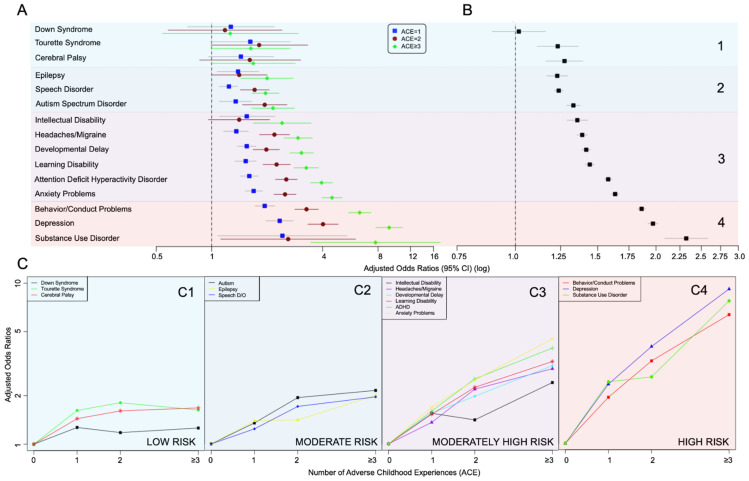
(**A**) Rank order of point estimate associations between household challenge ACEs and adjusted odds ratios of physical, developmental and behavioral health conditions surveyed in the National Survey of Child Health. (**B**) Associations between household challenge ACEs and odds ratios of behavioral and developmental health conditions surveyed in the National Survey of Child Health. (**C**) Dose-response natures of household challenge ACEs and odds ratio estimates.

**Figure 2 children-08-00761-f002:**
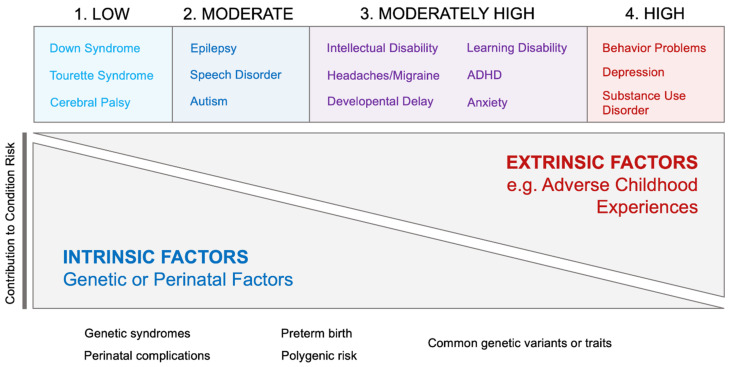
Model of how intrinsic and extrinsic factors may contribute to the risk of neurodevelopmental and behavioral health conditions.

**Table 1 children-08-00761-t001:** Household challenge adverse childhood experiences (ACEs) included in the ACE score.

Household Challenge ACEs
Child has ever experienced:
• Parental incarceration
• Family violence
• Household mental illness
• Household alcohol/drug problems
• Parental divorce/separation
• Parental death
• Household poverty

**Table 2 children-08-00761-t002:** Characteristics of the NSCH study population by ACE exposure, 2016–2019.

Variable	Number of Household Challenge ACEs				Chi-Sq. *p*-Value
	0	1	2	≥3	
No. of participants (N)	80,399	27,603	10,961	10,938	
Age of child (N, %)					
0–5 years	27,296 (33.95)	6471 (23.44)	1734 (15.82)	1313 (12.00)	<0.0001
6–11 years	23,970 (29.81)	8437 (30.57)	3462 (31.58)	3520 (32.18)	
≥12 years	29,133 (36.24)	12,695 (45.99)	5765 (52.60)	6105 (55.81)	
Sex (N, %)					
Male	41,609 (51.75)	14,332 (51.92)	5596 (51.05)	5566 (50.89)	0.15
Female	38,790 (48.25)	13,271 (48.08)	5365 (48.95)	5372 (49.11)	
Child’s Race/Ethnicity (N, %)					
Hispanic	8208 (10.21)	3620 (13.11)	1540 (14.05)	1494 (13.66)	<0.0001
Non-Hispanic White	57,846 (71.95)	18,447 (66.83)	7156 (65.29)	7161 (65.47)	
Non-Hispanic Black	3668 (4.56)	2317 (8.39)	1026 (9.36)	857 (7.84)	
Non-Hispanic Other	10,677 (13.28)	3219 (11.66)	1239 (11.30)	1426 (13.04)	
Parental Education (N, %)					
Less than high school	1413 (1.76)	799 (2.89)	340 (3.10)	425 (3.89)	<0.0001
High school	6898 (8.58)	4551 (16.49)	2342 (21.37)	2500 (22.86)	
College or higher	71,751 (89.24)	22,094 (80.04)	8221 (75.00)	7938 (72.57)	
Missing	337 (0.42)	159 (0.58)	58 (0.53)	75 (0.69)	
Family income to poverty ratio (N, %)					
<1.0	5314 (6.61)	3951 (14.31)	2231 (20.35)	2630 (24.04)	
1.0–1.9	9107 (11.33)	5909 (21.41)	2692 (24.56)	3011 (27.53)	<0.0001
2.0–3.9	23,898 (29.72)	9419 (34.12)	3559 (32.47)	3359 (30.71)	
≥4.0	42,080 (52.34)	8324 (30.16)	2479 (22.62)	1938 (17.72)	
Low birth weight child (N, %)					
Yes	5812 (7.23)	2242 (8.12)	1002 (9.14)	1037 (9.48)	<0.0001
No	71,289 (88.67)	23,992 (86.92)	9322 (85.05)	9065 (82.88)	
Missing	3298 (4.10)	1369 (4.96)	637 (5.81)	836 (7.64)	
Preterm birth (N, %)					
Yes	7912 (9.84)	3212 (11.64)	1381 (12.60)	1367 (12.50)	<0.0001
No	71,511 (88.95)	23,942 (86.74)	9367 (85.46)	9314 (85.15)	
Missing	976 (1.21)	449 (1.63)	213 (1.94)	257 (2.35)	
Household Smoking (N, %)					
Yes	7033 (8.75)	4947 (17.92)	2825 (25.77)	3840 (35.11)	<0.0001
No	72,903 (90.68)	22,507 (81.54)	8077 (73.69)	7032 (64.29)	
Missing	463 (0.58)	149 (0.54)	59 (0.54)	66 (0.60)	

NSCH, National Survey of Children’s Health; ACE, Adverse childhood experience.

**Table 3 children-08-00761-t003:** Odds ratio (95% CI) of health condition as a function of the number of ACEs.

Health Condition	Number of Household Challenge ACEs			
	0	1	2	≥3
Down Syndrome, (N = 129,533)				
Cases/total participants	126/80160	74/27539	26/10927	20/10907
Unadjusted Model	(reference)	1.42 (0.82–2.46)	1.36 (0.65–2.85)	1.58 (0.71–3.50)
Adjusted Model	(reference)	1.27 (0.74–2.19)	1.18 (0.58–2.41)	1.26 (0.54–2.95)
Tourette Syndrome, (N = 129,506)				
Cases/total participants	164/80187	85/27504	42/10925	52/10890
Unadjusted Model	(reference)	1.74 (1.11–2.74) *	2.09 (1.15–3.81) *	2.20 (1.37–3.55) **
Adjusted Model	(reference)	1.62 (0.98–2.67)	1.81 (0.99–3.32)	1.63 (0.99–2.67)
Cerebral Palsy, (N =129,400)				
Cases/total participants	188/80113	130/27492	51/10914	65/10881
Unadjusted Model	(reference)	1.63 (1.10–2.42) *	2.05 (1.14–3.71) *	2.30 (1.41–3.74) ***
Adjusted Model	(reference)	1.44 (0.96–2.17)	1.61 (0.86–3.04)	1.68 (0.99–2.84)
Epilepsy or Seizure Disorder, (N = 129,481)				
Cases/total participants	669/80143	357/27519	167/10920	200/10899
Unadjusted Model	(reference)	1.47 (1.16–1.86) **	1.61 (1.16–2.24) **	2.39 (1.79–3.17) ***
Adjusted Model	(reference)	1.39 (1.07–1.80) *	1.41 (1.00–2.00)	2.00 (1.44–2.77) ***
Speech Disorder, (N = 129,499)				
Cases/total participants	5340/80168	2370/27509	1132/10922	1303/10900
Unadjusted Model	(reference)	1.25 (1.13–1.39) ***	1.76 (1.49–2.08) ***	2.05 (1.76–2.38) ***
Adjusted Model	(reference)	1.24 (1.10–1.38) ***	1.71 (1.43–2.05) ***	1.96 (1.66–2.32) ***
Autism Spectrum Disorder, (N = 129,309)				
Cases/total participants	1522/80040	939/27483	502/10897	552/10889
Unadjusted Model	(reference)	1.57 (1.29–1.91) ***	2.46 (1.88–3.23) ***	2.98 (2.35–3.77) ***
Adjusted Model	(reference)	1.35 (1.10–1.65) **	1.94 (1.47–2.56) ***	2.15 (1.64–2.81) ***
Intellectual Disability, (N = 129,427)				
Cases/total participants	528/80106	360/27506	172/10918	269/10897
Unadjusted Model	(reference)	1.91 (1.39–2.64) ***	1.96 (1.37–2.80) ***	3.58 (2.59–4.96) ***
Adjusted Model	(reference)	1.55 (1.10–2.20) *	1.41 (0.96–2.06)	2.41 (1.68–3.47) ***
Severe or Frequent Headaches, (N = 129,504)				
Cases/total participants	2412/80164	1372/27509	874/10923	1189/10908
Unadjusted Model	(reference)	1.69 (1.45–1.96) ***	3.14 (2.62–3.77) ***	4.65 (3.93–5.49) ***
Adjusted Model	(reference)	1.36 (1.16–1.58) ***	2.19 (1.82–2.64) ***	2.94 (2.47–3.51) ***
Developmental Delay, (N = 129,413)				
Cases/total participants	3898/80123	2248/27496	1109/10906	1503/10888
Unadjusted Model	(reference)	1.64 (1.47–1.84) ***	2.17 (1.86–2.52) ***	3.42 (2.99–3.91) ***
Adjusted Model	(reference)	1.55 (1.38–1.74) ***	1.98 (1.68–2.33) ***	3.07 (2.65–3.56) ***
Learning Disability, (N = 112,531)				
Cases/total participants	3439/66996	2191/24675	1247/10318	1824/10542
Unadjusted Model	(reference)	1.76 (1.56–2.00) ***	2.83 (2.40–3.32) ***	4.35 (3.79–4.99) ***
Adjusted Model	(reference)	1.53 (1.34–1.74) ***	2.25 (1.90–2.67) ***	3.26 (2.80–3.80) ***
Attention Deficit Hyperactivity Disorder, (N = 128,897)				
Cases/total participants	5029/79840	3098/27365	1859/10855	2677/10837
Unadjusted Model	(reference)	1.81 (1.64–2.00) ***	3.16 (2.81–3.57) ***	5.36 (4.78–6.00) ***
Adjusted Model	(reference)	1.60 (1.43–1.79) ***	2.54 (2.22–2.91) ***	3.95 (3.44–4.53) ***
Anxiety Problems, (N = 129,488)				
Cases/total participants	5166/80183	3101/27498	1878/10920	2948/10887
Unadjusted Model	(reference)	1.82 (1.64–2.01) ***	2.92 (2.57–3.32) ***	5.63 (5.03–6.29) ***
Adjusted Model	(reference)	1.69 (1.52–1.89) ***	2.50 (2.18–2.87) ***	4.50 (3.96–5.12) ***
Behavior/Conduct Problems, (N = 129,509)				
Cases/total participants	3623/80185	2508/27505	1660/10918	2830/10901
Unadjusted Model	(reference)	2.14 (1.91–2.40) ***	3.91 (3.42–4.46) ***	7.98 (7.08–9.00) ***
Adjusted Model	(reference)	1.94 (1.72–2.20) ***	3.27 (2.83–3.79) ***	6.36 (5.53–7.32) ***
Depression, (N = 129,504)				
Cases/total participants	1704/80187	1477/27508	1065/10919	2058/10890
Unadjusted Model	(reference)	2.76 (2.34–3.25) ***	5.29 (4.42–6.34) ***	12.82 (11.06–14.87) ***
Adjusted Model	(reference)	2.34 (1.97–2.77) ***	4.02 (3.33–4.86) ***	9.19 (7.79–10.84) ***
Substance Use Disorder, (N = 92,132)				
Cases/total participants	53/52588	51/20900	46/9135	132/9509
Unadjusted Model	(reference)	2.85 (1.30–6.26) **	3.39 (1.61–7.15) **	12.13 (6.18–23.84) ***
Adjusted Model	(reference)	2.42 (1.08–5.44) *	2.60 (1.12–6.04) *	7.75 (3.45–17.39) ***

* *p* < 0.05, ** *p* < 0.01, *** *p* < 0.001.

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
