# Peer review of "Adverse Childhood Experiences Predict Common Neurodevelopmental and Behavioral Health Conditions among U.S. Children"

_children, 2021, doi:10.3390/children8090761_

Round 1
Reviewer 1 Report
Thank you for the opportunity to review this interesting and unique manuscript. This study took an innovative approach to delineating the relationship between ACEs associated with household dysfunction and several neurodevelopmental and behavioral health conditions. My suggestions/comments are:
- This study did not explore ACEs associated with maltreatment variables as they were not available. As such, the authors focus most of the discussion on non-maltreatment related topics. However, ACEs associated with maltreatment are among the strongest predictors of behavioral health conditions. Further, many ACEs associated with household dysfunction co-occur at high frequency with maltreatment ACEs. In other words, some of the results presented here, especially for the behavioral health conditions, may be at least partially driven by maltreatment ACEs. Though it is briefly mentioned in the discussion, I would like to see more discussion/acknowledgement about the relationship between maltreatment ACEs, household dysfunction ACEs, and behavioral health conditions.
- Please include a more detailed explanation for the 3 or more ACEs cut-off. Traditionally, 4+ ACEs is the cut-off used in the literature. I understand why it was not used here (i.e. not all ACEs were included, so the lower cut-off is justified), but it should be explained.
- In paragraph two of the discussion, the authors discuss their use of the dose-response effects of this study. It is also mentioned on line 313. I think this is one of the most interesting pieces of this study and sets it apart from other work. Please expand on why these dose-response effects are important.
- Please add several sentences that discuss the ways this research can be used by practitioners (i.e. physicians, psychologists, social workers, etc.). The readership of this journal leans towards the applied realm and implications to on-the-ground practice are worth exploring.
Again, this is excellent work! Thank you for your valuable contribution to the ACEs literature.
Author Response
Thank you so much for the insightful suggestions and comments. We agree with all the excellent points you have raised, and we made the additions to the manuscript as suggested. Below you will find your original comments (in black) and out responses to the individual points (in red). Thank you so much for the review and supportive comments.
Thank you for the opportunity to review this interesting and unique manuscript. This study took an innovative approach to delineating the relationship between ACEs associated with household dysfunction and several neurodevelopmental and behavioral health conditions. My suggestions/comments are:
1. This study did not explore ACEs associated with maltreatment variables as they were not available. As such, the authors focus most of the discussion on non-maltreatment related topics. However, ACEs associated with maltreatment are among the strongest predictors of behavioral health conditions. Further, many ACEs associated with household dysfunction co-occur at high frequency with maltreatment ACEs. In other words, some of the results presented here, especially for the behavioral health conditions, may be at least partially driven by maltreatment ACEs. Though it is briefly mentioned in the discussion, I would like to see more discussion/acknowledgement about the relationship between maltreatment ACEs, household dysfunction ACEs, and behavioral health conditions.
Thank you for the suggestion. We have expanded on these points in the discussion section (page 8 of the revised manuscript).
2. Please include a more detailed explanation for the 3 or more ACEs cut-off. Traditionally, 4+ ACEs is the cut-off used in the literature. I understand why it was not used here (i.e. not all ACEs were included, so the lower cut-off is justified), but it should be explained.
Thank you for the suggestion. We have provided a more detailed explanation about this point in the materials and methods section (page 3 of the revised manuscript, under “Statistical Analyses”).
3. In paragraph two of the discussion, the authors discuss their use of the dose-response effects of this study. It is also mentioned on line 313. I think this is one of the most interesting pieces of this study and sets it apart from other work. Please expand on why these dose-response effects are important.
Thank you for the suggestion. We have expanded on this point in the discussion section (page 7 of the revised manuscript) and added an additional figure to model this proposed relationship (Figure 2).
4. Please add several sentences that discuss the ways this research can be used by practitioners (i.e. physicians, psychologists, social workers, etc.). The readership of this journal leans towards the applied realm and implications to on-the-ground practice are worth exploring.
Thank you for the suggestion. We have expanded on this point in the discussion section (page 9 of the revised manuscript).
Again, this is excellent work! Thank you for your valuable contribution to the ACEs literature.

Reviewer 2 Report
I would like to congratulate the authors for the quality of their manuscript and its potential interest. I would like to make only a couple of suggestions that could help to improve their work:
First, the concept of "household challenges" is key to the article, but in my opinion it is not clearly explained. Some examples are provided (incarceration, violence) but it could be interesting to suggest a definition of the concept.
Second, the health conditions that are associated with ACEs are an interesting combination of internalizing and externalizing disorders. I think that could be interesting to further explore this in the discussion section (i.e suggesting differential etiological neurodevelopmental factors for internalizing and externalizing problems, or their association with specific household ACEs).
Third, the authors indicate that household challenges are not independent of other ACEs like child abuse. Could their results be attributable to child abuse (not evaluated in this research) that could be the real cause of the health conditions?
Author Response
Thank you so much for the insightful suggestions and comments. We agree with all the excellent points you have raised, and we made the additions to the manuscript as suggested. Below you will find your original comments (in black) and our responses to the individual points (in red). Thank you so much for the review and supportive comments.
I would like to congratulate the authors for the quality of their manuscript and its potential interest. I would like to make only a couple of suggestions that could help to improve their work:
First, the concept of "household challenges" is key to the article, but in my opinion it is not clearly explained. Some examples are provided (incarceration, violence) but it could be interesting to suggest a definition of the concept.
Thank you for the suggestion. We have provided a clearer explanation of household challenges in the introduction section (page 2 of the revised manuscript).
Second, the health conditions that are associated with ACEs are an interesting combination of internalizing and externalizing disorders. I think that could be interesting to further explore this in the discussion section (i.e suggesting differential etiological neurodevelopmental factors for internalizing and externalizing problems, or their association with specific household ACEs).
Thank you for the suggestion. We have elaborated on this point in the discussion section (page 8 of the revised manuscript).
Third, the authors indicate that household challenges are not independent of other ACEs like child abuse. Could their results be attributable to child abuse (not evaluated in this research) that could be the real cause of the health conditions?
Thank you for the suggestion. We have expanded on this point in the discussion section (page 8 of the revised manuscript).
